Predicting amyloid proteins using attention-based long short-term memory

Li Zhuowen 15221427798@163.com
Punan Branch of Renji Hospital, Shanghai Jiao Tong University , Shanghai , China
Nguyen Hoang
Electronic publication date: 2025 Feb 7
Publication date: 2025
Volume: 11
Electronic Location ID: e2660
Received 2024 Sep 19; Accepted 2024 Dec 23
Copyright: © 2025 Li
Copyright year: 2025
Copyright holder: Li
License: This is an open access article distributed under the terms of the Creative Commons Attribution License, which permits unrestricted use, distribution, reproduction and adaptation in any medium and for any purpose provided that it is properly attributed. For attribution, the original author(s), title, publication source (PeerJ Computer Science) and either DOI or URL of the article must be cited.
License URL: https://creativecommons.org/licenses/by/4.0/

Keywords: Amyloid, Transformers, Deep learning, LSTM, Alzheimer, Attention

Funding: The authors received no funding for this work.

==============================
Alzheimer’s disease (AD) is one of the genetically inherited neurodegenerative disorders that mostly occur when people get old. It can be recognized by severe memory impairment in the late stage, affecting cognitive function and general daily living. Reliable evidence confirms that the enhanced symptoms of AD are linked to the accumulation of amyloid proteins. The dense population of amyloid proteins forms insoluble fibrillar structures, causing significant pathological impacts in various tissues. Understanding amyloid protein’s mechanisms and identifying them at an early stage plays an essential role in treating AD as well as prevalent amyloid-related diseases. Recently, although several machine learning methods proposed for amyloid protein identification have shown promising results, most of them have not yet fully exploited the sequence information of the amyloid proteins. In this study, we develop a computational model for in silico identification of amyloid proteins using bidirectional long short-term memory in combination with an attention mechanism. In the testing phase, our findings showed that the model developed by our proposed method outperformed those developed by state-of-the-art methods with an area under the receiver operating characteristic curve of 0.9126.

Introduction

In many organs and tissues, amyloid proteins tend to flock to create insoluble aggregates of different sizes. These amyloid aggregates can continuously build up to produce intracellular protein inclusions or extracellular plaques, most notably as a component of disease processes. They are mostly made up of β-sheet structures and have a fibrillary shape when aggregated (Rambaran & Serpell, 2008). Disease-related amyloid proteins, found as plaques inside the central nervous system, include those involved in the pathological trigger or growth of Alzheimer’s disease (AD). Amyloid-beta (A β) plaques, neurofibrillary tangles, and brain shrinkage are the hallmarks of AD, a neurodegenerative condition. It causes dementia most often (Beach et al., 2012; Matsui et al., 2008) and has a major societal effect (Nichols et al., 2019; Matsui et al., 2008). However, since AD’s clinical symptoms might overlap with those of other illnesses, including frontotemporal lobar degeneration or late-onset mental disorders, making a clinical diagnosis of the disease can be difficult (Wang et al., 2025). These illnesses may occasionally coexist with AD and have comparable clinical signs and symptoms (Beach et al., 2012; Hampel et al., 2021; Wang et al., 2022). In experimental laboratories, histochemical dyes (e.g., Congo Red, Thioflavin T, etc.) are frequently used for pathological examination to investigate whether amyloid proteins are present in tissues. Besides, mass spectrometry can be used to identify amyoid proteins and classify their type of accumulation and patterns (Vrana et al., 2009). Since the structures and sizes of proteins building up amyloid fibrils are highly varied (López de la Paz et al., 2002), more advanced techniques are required to detect amyloid’s mark and its type in particular tissues. Among known modern assays, liquid chromatography-tandem mass spectrometry (LC-MS/MS) is considered the most effective, stable, and accurate one. However, as a high-level technique with expensive chemicals used, applying it requires large budgets, skilled experimenters, and a longer time. To accelerate the screening process and save budgets, computational biologists have developed various in silico models using current computer-aided advances. Several studies have utilized computational methods to predict the propensity for β-amyloid formation (Orlando et al., 2019), determine the aggregation-prone areas (APRs) of amyloid proteins (Maurer-Stroh et al., 2010; Thangakani et al., 2014), and identify amyloidogenicity (Palato et al., 2019). Aggregation Nucleation Prediction in Peptides and Proteins (ANuPP), by Prabakaran et al. (2021), was developed using ensemble learning. This tool was designed to flexibly identify putative APRs in peptides and proteins, overcoming the shortcomings of many existing models at the time. With the continuous growth of empirical data on pharmaceutical, chemical, and biological sequences (Mendez et al., 2018), coupled with advancements in artificial intelligence (AI), there has been a growing focus over the past several decades on developing more efficient models for amyloid proteins, particularly leveraging bio-molecular representations in bio-cheminformatics (Nguyen-Vo et al., 2024; Chen et al., 2024). Advanced AI techniques have been utilized to analyze neuroimaging data and patient histories (Chen et al., 2017), aiding in the diagnosis and treatment of various neurological diseases (Pham et al., 2019). Attention mechanisms have been leveraged to improve model interpretability by focusing on the most relevant features in imaging data (Chen et al., 2021; Wu et al., 2024). Furthermore, hybrid models integrating recurrent neural networks (RNN) with attention mechanisms have been developed to capture temporal dynamics and essential features in patient data, resulting in enhanced prediction accuracy for AD-related outcomes (Li & Ma, 2022).

Related work

For years, many machine learning models have constructed to predict sequences using different representations, such as amino acid composition (AAC) (Bhasin & Raghava, 2004), composition transition distribution (CTDC) based on the percentage of particular amino acid property groups (Li et al., 2006; Dubchak et al., 1995; Chen et al., 2018), amphiphilic pseudo-amino acid composition (APAAC) (Chou, 2011), composition transition distribution difference (CTDD) based on distribution of amino acid properties in sequences (Chiti & Dobson, 2006; Vrana et al., 2009), and Dipeptide deviation from expected mean (DDE) (Saravanan & Gautham, 2015; Wang et al., 2020). RFAmyloid, developed by Niu et al. (2018), is one of the pioneered model designed to distinguish amyloid proteins from ordinary ones. It was created using random forest and different encoding schemes for sequence information extraction. They collected and curated protein sequence samples to obtain a dataset with 165 amyloid and 382 non-amyloid proteins for modeling. iAMY-SCM, by Charoenkwan et al. (2021), was developed the Scoring Card method to predict and analyze amyloid proteins. In their proposed method, a streamlined weighted-sum function was used in combination with thr propensity scores computed for dipeptides. iAMY-SCM was reported to achieve as good performance as RFAmyloid did based on metrics evaluated in cross-validation and independent testing (Charoenkwan et al., 2021). Charoenkwan et al. (2022) proposed AMYPred-FRL, an ensemble model with better performance reported. They combined six common machine learning algorithms, including logistic regression, k-nearest neighbors (k-NN), support vector machine (SVM), maximum gradient boosting (MGB), extremely randomized trees, and random forest (RF), and ten different sequence-based feature descriptors to construct 60 base models. In prediction, these base models returned 60 probabilistic features, which were then fed to the final meta-model (Charoenkwan et al., 2022). Most recently, Yang, Liu & Zhang (2023) proposed ECAmyloid, another ensemble machine learning model that detects amyloid proteins using diverse sequence-derived features. The sequence-based features combined information on the composition, evolution, and structure of the sequences. To select the most suitable models for ensemble learning, they used an enhanced classification selection method. The prediction outcomes of the meta-learner were based on the decision of all voters (base models) (Yang, Liu & Zhang, 2023).

Materials and Methods

Dataset

We used a refined dataset collected from Niu et al. (2018) for model development and evaluation, consisting of 165 amyloid proteins (positive samples) and 382 non-amyloid proteins (negative samples). The raw data was used without any transformations prior to the main analysis. The dataset was then randomly sampled to create two sets of data: a training set and an independent test set for modeling and performance assessment. The training set contains 132 positive samples and 305 negative samples, while the test set contains 33 positive samples and 77 negative samples. From the training set, we created a validation set to find the optimal point for our model. Table 1 provides information of sampled data in the training, validation, and test sets.

Table 1 Information on datasets used for model training, validation, and testing.

Dataset	Training	Validation	Test	
Positive	117	15	33	
Negative	276	29	77	
Total	393	44	110	

Justification for model type used

Existing computational approaches for amyloid protein identification each have their own strengths and weaknesses, highlighting the need for more effective methods. In this study, we propose an attention-based bidirectional long short-term memory (Bi-LSTM) model, which offers significant advantages over traditional models. The Bi-LSTM architecture is specifically chosen for its ability to capture bidirectional dependencies in sequence data, crucial for understanding the structural information of proteins (Nguyen-Vo et al., 2022). Unlike traditional LSTMs, which process data in a single direction and capture only past context, Bi-LSTMs process sequences in both forward and backward directions, allowing the model to utilize preceding and succeeding information. By integrating an attention layer, the model can selectively focus on important features, enhancing its accuracy for amyloid protein prediction. This approach addresses limitations in existing methods by leveraging the detailed information contained within protein structures.

Selection method

To develop and evaluate the proposed model, we utilized a refined dataset collected from Niu et al. (2018). The model’s architecture was carefully selected and optimized based on its validation performance on this dataset. We benchmarked the proposed model against other machine learning models trained on multiple commonly used feature sets, such as amino acid composition (AAC) (Wei et al., 2018), amino acid pair composition (APAAC) (Chen, Cheong & Siu, 2021), composition transition distribution (CTDC) (Zhao et al., 2022), composition transition distribution difference (CTDD) (Thi Phan et al., 2022), and dynamic positional encoding (DPE) (Li & Ma, 2022), to ensure a comprehensive assessment. AAC and APAAC provide foundational compositional insights, while CTDC and CTDD offer more detailed spatial information. DPE introduces a dynamic approach to positional encoding, making it suitable for advanced machine learning applications. Each method has its strengths and is selected based on the specific requirements of studies. Furthermore, we compared our approach with current computational frameworks for amyloid protein identification, including ECAmyloid (Yang, Liu & Zhang, 2023), AMYPred-FRL (Charoenkwan et al., 2022), iAMY-SCM (Charoenkwan et al., 2021), and RFAmyloid (Niu et al., 2018). This comparative analysis helped in selecting and validating the Bi-LSTM model with attention as the most effective architecture for our study.

Assessment metrics

The area under the receiver operating characteristic curve (AUROC) was chosen as the primary metric for model assessment due to its ability to evaluate the overall performance of the model across all possible classification thresholds, making it a robust indicator of discriminative ability. Additionally, balanced accuracy (BA), Matthews correlation coefficient (MCC), and F1 score (F1) were computed at the default threshold of 0.5 to provide a comprehensive evaluation of the model’s performance. BA accounts for imbalances in the dataset by considering both sensitivity and specificity, MCC provides a balanced measure of model quality even with imbalanced classes, and F1 score offers insight into the trade-off between precision and recall. Together, these metrics offer a well-rounded assessment of the model’s performance under various aspects of classification accuracy.

Other benchmarking models

To fairly assess the performance of our model, we compared it to a series of machine learning models developed using commonly used feature sets. We selected five learning algorithms, including k-NN (Fix & Hodges, 1989), Logistic Regression (LR) (Tolles & Meurer, 2016), SVM (Cortes & Vapnik, 1995), RF (Breiman, 2001), and eXtreme Gradient Boosting (XGB) (Chen & Guestrin, 2016), combined with five feature sets comprising AAC (Bhasin & Raghava, 2004), APAAC (Chou, 2011), CTDC (Li et al., 2006; Dubchak et al., 1995; Chen et al., 2018), CTDD (Chiti & Dobson, 2006; Vrana et al., 2009), and DPE (Saravanan & Gautham, 2015; Wang et al., 2020) to create 25 machine learning models. We trained these models using the same training set as used for training our model. The hyperparameters of these models were determined based on the validation set. Besides machine learning models, we also implemented several baseline deep learning models, including RNN and gated recurrent units (GRU), both with and without the integration of an attention (Att) mechanism.

To further assess the effectiveness of our model in practice, we compared it with other existing computational frameworks, including RFAmyloid (Niu et al., 2018), iAMY-SCM (Charoenkwan et al., 2021), AMYPred-FRL (Charoenkwan et al., 2022), and ECAmyloid (Yang, Liu & Zhang, 2023). For the RFAmyloid and iAMY-SCM frameworks, we accessed their web servers and uploaded our test samples to run the prediction tasks and then collected the results. Since these web servers predicted the classes of the samples (either amyloid protein or non-amyloid protein), we could only calculate balanced accuracy, MCC, and F1 score. For the AMYPred-FRL and ECAmyloid frameworks, we reimplemented them using the source codes provided by the authors and evaluated them on the same independent test set to report the results.

Our proposed method

Bidirectional long short-term memory

Long short-term memory (LSTM), belonging to the family of RNNs, is designed to tackle the problem of gradient vanishing when the models are forced to learn long-distance sequential data (Hochreiter & Schmidhuber, 1997). To address this problem, gating mechanisms (forget, input, and output gates) and memory cells are employed to control flow of information. LSTM can maintain and update its cell state over long periods. LSTM’s design helps it effectively deal with many tasks from language processing to time series prediction. The mathematical expression of the Forget gate is as follows:

(1) ft=δ(Wforget[ht−1,Xt]+bforget),

where δ denote the sigmoid activation function, Xt is the input vector at timestep t, and ht−1 is the hidden vector from the previous timestep t−1. The weights Wf and bias values bf are randomly initialized and gradually updated when the model learns. The mathematical expression of Input and Output gates are defined as:

(2) it=δ(Winput[ht−1,Xt]+binput),

(3) Ot=δ(Woutput[ht−1,Xt]+boutput).

The candidate memory cell ( Lt) contributes to adjusting the information in the current cell state ( Ct). The output Ot and current cell state ( Ct) are used to create the current hidden stage ( ht), as follows:

(4) Lt=tanh⁡(Wcell[ht−1,Xt]+bcell),

(5) Ct=ft×Ct−1+it×Lt,

(6) ht=Ot×tanh⁡(Ct).

Attention layer

Our attention layer accept hidden states H=[h1,h2,h3,⋯,hn] returned from the LSTM layer. The vectors of hidden states hk are then nonlinearly transformed to create activated output U=[u1,u2,u3,⋯,un] using the Tanh function:

(7) uk=tanh⁡(Wkhk+bk),

where Wk denotes the weight matrices and bk refers to the offset quantity at timestep k. Certain operational factors during the shield tunneling process have a significant impact on the direction and location of the shield tunnel, thus they should be given additional consideration. The attention mechanism is applied to create the attention weight matrix αk=[α1,α2,α3,α4⋯,αn]. This matrix stores important information of all single intermediate states, described as:

(8) αk=exp⁡(ukTus)∑k=1nexp(ukTus),

where αk refers to the normalized attention weight at timestep k and us is the time series attention matrix which are randomly initialized. Finally, for each input sample, a computed weighted sum of hidden states is obtained to create an attention vector V for the next stage:

(9) V=∑knαkhk.

Model architecture

The protein sequences, each represented as a distinct chain of letters, are encoded by the Embedding layer to form a sequence vector X=(X1,X2,X3,X4,…,Xt). First, the sequence vector length L is embedded after passing through the Embedding layer. The Embedding layer is defined by a vocabulary of 21 letters and an embedding dimension of 100. The embedding matrices, with dimensions L×100, are then input to the Bi-LSTM layer. After processing through the Bi-LSTM layer, the hidden dimension vectors at all timesteps are used to compute the attention scores using the Softmax function. The attention outputs, with dimensions 1×128, are then passed through the first Fully Connected (FC) layer, normalized using 1-Dimensional Batch Normalization (BatchNorm1D) (Ioffe & Szegedy, 2015), and activated by the rectified linear unit (ReLU) function. Finally, the activated outputs are transferred to the second FC layer, where they are activated by the Sigmoid function to produce the final prediction outcomes. Figure 1 visualizes the architecture of the proposed model.

Figure 1 Architecture of the proposed model.

Experimental settings

All the experiments were performed using a computer running Windows 11 Home, with an Intel Core i7-13700F CPU (16 cores/24 threads, max turbo 5.2 GHz). The computer is equipped with 2 × 16 GB RAM for temporary memory. All deep learning models were trained using the PyTorch platform (version 2) with CUDA Toolkit version 11.7. All libraries are compatible with Python 3.11.

We trained our models with the training set and monitored the modeling phase to find the optimal epoch for the model based on the validation losses. The optimization process was controlled using the Adam optimizer (Kingma & Ba, 2014). A learning rate of 0.001 was chosen for the entire modeling phase. Binary cross-entropy was used as the loss function for the training process. The model was trained over 40 epochs, and the loss function was found to converge at epoch 23.

Results and discussion

Table 2 summarizes the performance of all implemented models, including model developed using our methods and those developed with other machine learning ones.

Table 2 Performance of our model against other machine learning models.

Model	Features	AUROC	BA	MCC	F1	
	AAC	0.7627	0.6385	0.4073	0.3386	
	APAAC	0.8322	0.7403	0.4805	0.4805	
k-NN	CTDC	0.7470	0.5887	0.252	0.2147	
	CTDD	0.7227	0.6926	0.4884	0.4472	
	DDE	0.8213	0.7186	0.4372	0.4372	
	AAC	0.7052	0.5000	0	0	
	APAAC	0.8304	0.7359	0.4577	0.4561	
LR	CTDC	0.6348	0.5238	0.1340	0.0643	
	CTDD	0.7879	0.7684	0.5416	0.5415	
	DDE	0.8855	0.7965	0.5621	0.5547	
	AAC	0.8386	0.7792	0.5946	0.5890	
	APAAC	0.8351	0.6926	0.4103	0.4064	
RF	CTDC	0.7737	0.6840	0.4024	0.3953	
	CTDD	0.8996	0.8225	0.6256	0.6234	
	DDE	0.8579	0.7229	0.4638	0.4619	
	AAC	0.8520	0.7597	0.5345	0.5333	
	APAAC	0.8170	0.7424	0.4735	0.4726	
SVM	CTDC	0.7218	0.6299	0.3086	0.2927	
	CTDD	0.7696	0.7013	0.4881	0.4581	
	DDE	0.8467	0.7489	0.4898	0.4894	
	AAC	0.8142	0.7684	0.5416	0.5415	
	APAAC	0.8339	0.7165	0.4454	0.4444	
XGB	CTDC	0.7859	0.6688	0.3747	0.3662	
	CTDD	0.8863	0.8052	0.6104	0.6104	
	DDE	0.8819	0.7597	0.5345	0.5333	
	RNN	0.8930	0.8455	0.6236	0.6222	
	RNN-Att	0.8914	0.8455	0.6290	0.6288	
Deep learning	GRU	0.8874	0.8636	0.6682	0.6667	
	GRU-Att	0.8937	0.8818	0.7212	0.7210	
	Ours	0.9126	0.8788	0.7454	0.7447	

It can be observed that XGB is a promising model that can effectively learn features from protein sequences, as its metric values are remarkably greater than those evaluated for k-NN and SVM models. The AUROC values of the XGB model trained with the DDE and CTDD feature sets are 0.8819 and 0.8863, respectively. Additionally, the RF model trained with the DDE feature set has higher predictive power than models trained with the other feature sets, with an AUROC value of 0.8855. Compared to these machine learning models, our model achieves the highest performance with an AUROC value of 0.9126. The balanced accuracy and F1 score of our model also outperform the other models, with values of 0.8788 and 0.7447, respectively.

The comparative analysis between our model and the other computational frameworks is shown in Table 3. The results indicate that the model developed using our proposed method has better performance compared to the existing frameworks, with an AUROC value of 0.9126 and a balanced accuracy of 0.8788. Regarding the MCC, our model remains the best-performing model with a value of 0.7454. Our model’s F1 score is also greater than that of other models, with a value of 0.7447.

Table 3 Performance of our model against other computational frameworks.

Model	AUROC	BA	MCC	F1	
RFAmyloid	–	0.8364	0.6104	0.6104	
iAMY-SCM	–	0.8182	0.5600	0.5595	
AMYPred-FRL	0.9055	0.8636	0.6682	0.6667	
ECAmyloid	0.8937	0.8455	0.6236	0.6222	
Ours	0.9126	0.8788	0.7454	0.7447	

Model’s robustness examination

The performance of our model was evaluated over 10 random modeling trials, as summarized in Table 4. Across these trials, the model demonstrates robust performance, with an average AUROC value of 0.9167, indicating a high capability for distinguishing between classes. Similarly, the mean balanced accuracy of 0.8898 reflects a strong balance in sensitivity and specificity. The average MCC value of 0.7447 further confirms the reliability of the model in terms of binary classification correlation. Finally, the averaged F1 score of 0.7438 highlights its consistent precision and recall balance. These results demonstrate the stability and efficacy of the model, as evidenced by the low standard deviations across all metrics.

Table 4 Performance of our model over 10 random modeling trials.

Trial	AUROC	BA	MCC	F1	
1	0.9126	0.8788	0.7454	0.7447	
2	0.9104	0.8846	0.7386	0.7379	
3	0.9297	0.8942	0.7469	0.7451	
4	0.9094	0.8846	0.7300	0.7293	
5	0.9356	0.9038	0.7667	0.7659	
6	0.9036	0.8846	0.7251	0.7244	
7	0.9320	0.8942	0.7515	0.7498	
8	0.9097	0.8846	0.7345	0.7338	
9	0.9163	0.9038	0.7781	0.7781	
10	0.9072	0.8846	0.7300	0.7293	
Mean	0.9167	0.8898	0.7447	0.7438	
SD	0.0115	0.0087	0.0170	0.0171	

Limitations and future work

The proposed method, while demonstrating strong performance in amyloid protein identification using Bi-LSTM combined with an attention mechanism, does have certain drawbacks. One primary concern is the model’s reliance on sequence information alone. Although this approach captures essential patterns within protein sequences, it may overlook critical structural and functional information that could enhance predictive accuracy. For instance, complex folding processes often occur in amyloid proteins, and their 3D structures and interactions with other biomolecules closely influence their pathological effects. By focusing solely on sequence data, the model could miss these aspects, potentially limiting its generalizability to different types of amyloid proteins or other related neurodegenerative diseases.

Another limitation is the potential risk of overfitting, particularly given the complexity of the Bi-LSTM and attention mechanisms used. Overfitting occurs when a model learns to perform exceptionally well on the training data but fails to generalize to new, unseen data. If the training dataset is not sufficiently large or diverse, it exacerbates this risk. Moreover, the computational complexity of the model can pose practical challenges, especially when dealing with large-scale datasets or deploying it in real-time diagnostic settings. The interpretability of the model’s decisions is also a concern, as the intricate layers of Bi-LSTM and attention mechanisms can obscure the reasoning behind predictions, making it difficult for researchers or clinicians to trust and verify the model’s outputs.

For future work, it would be beneficial to explore the integration of additional types of data, such as structural information from 3D protein folding patterns or interaction data with other biomolecules. This multi-modal approach could provide a more comprehensive understanding of amyloid proteins and potentially improve the model’s accuracy and generalizability. Another promising direction is the development of techniques to enhance model interpretability, such as attention visualization or layer-wise relevance propagation, which would allow researchers to better understand which parts of the protein sequence contribute most to the predictions. Additionally, efforts should be made to reduce the model’s computational demands, possibly through model optimization or the use of more efficient architectures, to facilitate its deployment in practical, real-world scenarios. Finally, ongoing validation with larger and more diverse datasets will be crucial to ensuring the model’s robustness and applicability across different populations and conditions.

Conclusion

In this work, we presented an effective model for the in silico screening of amyloid proteins using attention-based Bi-LSTM. The introduction of an attention layer helps the model selectively focus on essential information and extract important features from amyloid protein sequences. Experimental results indicated that the model developed using our proposed method improved performance on the independent test set compared to other machine learning models as well as existing computational frameworks. Other metrics, including balanced accuracy, MCC, and F1 score, also demonstrated the robustness of the proposed method. This method can be modified to suit various dataset sizes and applied to address similar problems.

Supplemental Information

Supplemental Information 1 The Python code and data used in the experiments.

Supplemental Information 2 An introduction and explanation of the code, and steps for implementation.

Supplemental Information 3 Raw data results from all baseline machine learning models.

Supplemental Information 4 Raw data results from all comparing models.

Additional Information and Declarations

Competing Interests

The authors declare that they have no competing interests.

Author Contributions

Zhuowen Li conceived and designed the experiments, performed the experiments, analyzed the data, performed the computation work, prepared figures and/or tables, authored or reviewed drafts of the article, and approved the final draft.

Data Availability

The following information was supplied regarding data availability:

The Python code and data used in the experiments are available in the Supplemental File.

The data from Niu et al. (2018) is available at GitHub: https://github.com/KOALA-L/ECAmyloid/tree/main/dataset.

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
