# Peer review of "Predicting amyloid proteins using attention-based long short-term memory"

_PeerJ Computer Science, doi:10.7717/peerj-cs.2660_

## Round 0.1 · original submission · Major Revisions

Based on reviewers' comments, we have found that your work need to be significantly improved. However, we are happy to offer you an opportunity to revise your work. Please check and address all points raised by reviewers.

Reviewer 1 ·

Basic reporting

The author presents a model based on LSTM with an attention mechanism to predict Amyloid proteins. Overall, the work is sound, the experiments are well-designed, and the paper presents itself in a high standard of English. I recommend this work for publication once the following issues are addressed.

Experimental design

(1)  Line 102: “… commonly used feature sets, such as AAC, APAAC, CTDC, CTDD, and DPE, …”. You are recommended to explain one by one how these encoding schemes are different from each other and highlight specific characteristics of each type. 

(2) 
- Line 102: “… commonly used feature sets, such as AAC, APAAC, CTDC, CTDD, and DPE, …”.
- Lines 123 – 126: “combined with five feature sets comprising 124 AAC (Bhasin and Raghava, 2004), APAAC (Chou, 2011), CTDC (Li et al., 2006; Dubchak et al., 1995; 125 Chen et al., 2018), CTDD (Chiti and Dobson, 2006; Vrana et al., 2009b), and DDE (Saravanan and 126 Gautham, 2015; Wang et al., 2020)” 
Are “CTDD” and “CTDC” the same?
Are “DDE” and “DPE” the same?
Can you check and clarify it? It looks quite confusing.

(3) Line 184: “We trained our model with the training set …”, If you implement more than one deep learning model, “model” must be in plural form (“models”). Please correct.

(4) Grammar/Text/Typo mistakes.
- Line 54: “RELATE WORK” should be “RELATED WORKS”
- Table 2: “k-NN”. You can use KNN or k-NN but you shouldn’t use two different types of abbreviations. Select one of them only.

(5) Are there any strengths (e.g., speed) that make your work a better solution compared to other works, aside from performance?

Validity of the findings

Have all models been re-implemented multiple times to provide statistical evidence of the model's reproducibility? If not, please perform additional experiments using different data sampling seeds, develop the corresponding models, and evaluate them.

Cite this review as

Reviewer 2 ·

Basic reporting

- Summary: This paper proposed attention-based LSTM method for predicting Amyloid protein to support Alzheimer's disease (AD) treatment. Previous methods haven’t yet fully exploited the sequence information of the amyloid proteins; therefore, this paper focuses on it to understand amyloid protein’s mechanisms and identify them at an early stage plays an essential role in treating AD as well as prevalent amyloid-related diseases. The authors evaluated their proposed method to show their superior performance when compared to state-of-the-art methods.

Experimental design

- Technical novelty and soundness:
+ The proposed method is straightforward and easy to understand. The motivation of the proposed method comes from the LSTM that can handle sequential data.
+ The authors proposed “Bidirectional LSTM”, but there is no detail or not clear about the “Bidirectional”. The authors should revise and clearly explain it.

Validity of the findings

- Experiment:
+ The authors should compare with other learning-based methods for a fair comparison together with the current methods.
+ The authors should consider evaluating the proposed method on more datasets to fully evaluate their performance. Because this is a real-world problem, and there will be many varieties of patient’s disease which will be different from the one in the current dataset.

- Writing and presentation:
+ The paper is structured with enough components, including introduction, related works, method, experiments, limitations and future work, and conclusion.
+ I suggest the equations should be aligned in center of the line. Figure 1 should be converted to horizontal as now, it is wasting of space.
+ The authors should discuss more about the related works, such as attention architecture, LSTM, then previous methods on applying different techniques to medical field or for aiding AD treatment.

Cite this review as

Reviewer 3 ·

Basic reporting

This work aims to build a computational model to predict amyloid proteins using attention-based deep learning models. The paper has merits, such as a well-organized structure, professional language, and providing sufficient background for understanding. However, there are points that need revision:
- Some sentences in the Introduction (e.g., lines 23–33) and Discussion (e.g., lines 218–230) are unclear. The authors should proofread and revise these sections, and where possible, add transition words to improve clarity and overall readability.
- The manuscript should incorporate more recent studies, particularly those focusing on deep learning-based approaches for sequence analysis.
- The section on related works (lines 54–78) does not sufficiently explain why Bi-LSTM was chosen over newer models. A comparison with more modern deep learning techniques would provide stronger justification for selecting this model.

Experimental design

- The paper does not address how the relatively small dataset used might impact model performance, nor does it mention potential class imbalance issues. These should be discussed. Information on preprocessing (lines 82–83) is missing.
- The authors should either justify why preprocessing was omitted or explore its effects.
- The study does not compare the Bi-LSTM model with other deep learning architectures, such as CNNs, transformers, or GRUs. Without these comparisons, it is difficult to assess the model's effectiveness. Including such comparisons would significantly strengthen the analysis.

Validity of the findings

- There is no statistical evidence provided in the paper. This issue should be addressed to support the findings.

Cite this review as

---

## Round 0.2 · accepted · Accept

Based on the reviewer's comments, we are pleased to inform you that your work has been accepted for publication at PeerJ Computer Science.

Reviewer 1 ·

Basic reporting

No comment

Experimental design

No comment

Validity of the findings

No comment

Additional comments

The authors have fixed all the issues pointed out in my previous report. They have performed more experiments to support the findings. The revised manuscript is now much better, and it can be accepted for publication.

Cite this review as

Reviewer 3 ·

Basic reporting

- The manuscript is clear and well-written, with the introduction and discussion sections significantly improved for clarity.
- Relevant recent studies have been added, providing a better background and meeting the scope of the journal.

Experimental design

- The Bi-LSTM model is well-justified, with added comparisons to other models, enhancing the robustness of the study.
- The authors have addressed concerns about dataset size, preprocessing, and class imbalance, aligning with the journal's standard.

Validity of the findings

- Statistical analyses, including results from multiple trials, confirm the robustness and reliability of the findings.
- The model’s performance on key metrics (AUROC, F1 score) aligns with the journal’s scope on valid results.

Cite this review as